

# Expression of vitamin D 1α-hydroxylase in human gingival fibroblasts in vivo

Kaining Liu[1,2,3,4], Bing Han[2,3,4,5], Jianxia Hou[1,2,3,4], Jianyun Zhang[2,3,4,6], Jing Su[7] and Huanxin Meng[1,2,3,4]

[1] Department of Periodontology, Peking University School and Hospital of Stomatology, Beijing, China
[2] National Clinical Research Center for Oral Diseases, Beijing, China
[3] National Engineering Laboratory for Digital and Material Technology of Stomatology, Beijing, China
[4] Beijing Key Laboratory of Digital Stomatology, Beijing, China
[5] Department of Cariology and Endodontology, Peking University School and Hospital of Stomatology, Beijing, China
[6] Department of Oral Pathology, Peking University School and Hospital of Stomatology, Beijing, China
[7] Department of Pathology, Peking University Health Science Center, Beijing, China

## ABSTRACT

**Background.** Vitamin D 1α-hydroxylase CYP27B1 is the key factor in the vitamin D pathway. Previously, we analyzed the expression of CYP27B1 in human gingival fibroblasts in vitro. In the present study, we analyzed the gingival expression of CYP27B1 in vivo.

**Methods.** Forty-two patients with periodontitis Stage IV Grade C and 33 controls were recruited. All patients with periodontitis had unsalvageable teeth and part of the wall of the periodontal pocket was resected and obtained after tooth extraction. All controls needed crown-lengthening surgery, and samples of gingiva resected during surgery were also harvested. All the individuals' gingivae were used for immunohistochemistry and immunofluorescence. In addition, gingivae from seventeen subjects of the diseased group and twelve subjects of the control group were analyzed by real-time PCR.

**Results.** Expression of CYP27B1 was detected both in gingival epithelia and in gingival connective tissues, and the expression in connective tissues colocalized with vimentin, indicating that CYP27B1 protein is expressed in gingival fibroblasts. The expression of CYP27B1 mRNA in gingival connective tissues and the CYP27B1 staining scores in gingival fibroblasts in the diseased group were significantly higher than those in the control group.

**Conclusions.** Expression of CYP27B1 in human gingival tissues was detected, not only in the fibroblasts of gingival connective tissues, but also in the gingival epithelial cells, and might be positively correlated with periodontal inflammation.

# INTRODUCTION

Vitamin $D_3$ is of great importance in regulating calcium and phosphorus metabolism and immunological responses (*Sassi, Tamone & D'Amelio, 2018*; *Medrano et al., 2018*). The active hormonal metabolite of vitamin $D_3$, $1α,25$-dihydroxyvitamin $D_3(1,25OH_2D_3)$, is formed by two-step hydroxylations (*Medrano et al., 2018*): (1) from vitamin $D_3$ to

Corresponding author
Huanxin Meng,
kqhxmeng@bjmu.edu.cn

25-hydroxyvitamin $D_3$ (25OHD$_3$) by vitamin D 25-hydroxylase in the liver, followed by (2) from 25OHD$_3$ to 1,25OH$_2$D$_3$ by vitamin D 1 $\alpha$-hydroxylase in the kidney. Vitamin D 1 $\alpha$-hydroxylase CYP27B1 was first detected in the kidney (*Takeyama et al., 1997*), but subsequently extra-renal sites of 1,25OH$_2$D$_3$ synthesis were also verified, including the skin (*Bikle & Christakos, 2020*), prostate (*Capiod et al., 2018*), bone (*Van Driel et al., 2006*), eye (*Alsalem et al., 2014*; *Markiewicz et al., 2019*), blood vessels (*Somjen et al., 2005*; *Zehnder et al., 2002*), human periodontal ligament cells and human gingival fibroblasts (hGFs) (*Liu, Meng & Hou, 2012a*; *Liu, Meng & Hou, 2012b*).

The vitamin D pathway, including connected reactions from the activation of toll-like receptors to the expression of the human cationic antimicrobial protein of 18 kDa (hCAP18) in monocytes, was first proposed in 2006 (*Liu et al., 2006*). hCAP18 is the precursor of the important antimicrobial peptide, cathelicidin (composed of 37 amino acids, also called LL37), which is the end product of the vitamin D pathway (*Liu et al., 2006*). LL37 has a broad-spectrum antibacterial effect, and has a regulatory effect on the immuno-inflammatory response (*Teles et al., 2013*; *Xhindoli et al., 2016*; *Wang et al., 2019*). A similar pathway also exists in keratinocytes (*Schauber et al., 2007*). In our previous study (*Gao, Liu & Meng, 2018*), the vitamin D pathway was detected in hGFs, and vitamin D 1 $\alpha$-hydroxylase CYP27B1 was demonstrated to be the key factor in the pathway. Our results suggested that the vitamin D pathway might be important in periodontal immune defense (*Gao, Liu & Meng, 2018*), which was in line with another research group (*Zhou et al., 2018*). As the key factor in the vitamin D pathway (*Gao, Liu & Meng, 2018*), CYP27B1 is worthy of further research. To our knowledge, however, the in vivo expression of CYP27B1 in hGFs has not been reported.

Although CYP27B1 expressed in hGFs in vitro is the same as that in kidney, its regulation is different: periodontitis-related inflammatory stimuli interleukin-1$\beta$ (IL-1$\beta$), sodium butyrate and *Porphyromonas gingivalis* lipopolysaccharide (*Pg*-LPS) induce significant up-regulation of CYP27B1, while regulators of 1 $\alpha$-hydroxylase in kidney (parathyroid hormone, calcium and 1,25OH$_2$D$_3$) do not significantly influence the expression of CYP27B1 in hGFs in vitro (*Gao, Liu & Meng, 2018*; *Liu, Meng & Hou, 2012b*). However, the actual situation in vivo is much more complicated than that simulated in vitro. Previously, our group reported that IL-1$\beta$ and butyric acid, which are both up-regulators of CYP27B1, could be detected in the gingival crevicular fluids of patients with periodontitis, and the concentrations were positively correlated with periodontal inflammation (*Liu et al., 2010*; *Lu et al., 2014*). Thus, based on our previous data, it might be hypothesized that CYP27B1 is expressed in hGFs in vivo and patients with periodontitis might have stronger expression. The aim of this study was to test the above hypothesis and to elucidate the features of CYP27B1 expression in hGFs in vivo. In addition, the expression of CYP27B1 in human gingival epithelial cells (hGEs) was analyzed.

## MATERIALS & METHODS

### Tissue sampling

The institutional review board of Peking University School and Hospital of Stomatology approved the study protocol (PKUSSIRB-2011007). Written informed consent was obtained from each participant in accordance with the Declaration of Helsinki.

Forty-two patients with periodontitis Stage IV Grade C and 33 healthy controls were enrolled from the clinic of the Periodontology Department, Peking University School and Hospital of Stomatology. On the basis of the 2017 World Workshop on the Classification of Periodontal and Peri-Implant Diseases and Conditions (*Tonetti, Greenwell & Kornman, 2018*; *Lang & Bartold, 2018*; *Papapanou et al., 2018*), diagnosis was made for each individual after complete periodontal examination. The inclusion criteria were as follows. Periodontitis: at least eight teeth with probing depth (PD) $\geq$ 7 mm and evidence of alveolar bone loss on radiographs; at least four teeth with mobility II or III; at least one unsalvageable tooth with mobility III and alveolar bone resorption close to the root apex, needing to be extracted. Healthy controls: no site with attachment loss (AL); no site with PD > 3 mm after supragingival scaling; no radiographic evidence of alveolar bone loss; less than 10% of sites with bleeding on probing (BOP); at least one tooth needing crown-lengthening surgery. Any subjects with systemic diseases or pregnancy were excluded. All 75 subjects enrolled were non-smokers.

The PD of all the enrolled subjects' teeth and AL of each unsalvageable tooth or each control tooth needing crown-lengthening surgery were recorded at six sites (mesial, distal, and middle sites of facial and lingual sides). Bleeding index (BI) (*Mazza, Newman & Sims, 1981*) was also recorded for each tooth of each individual. The mean PD, AL and BI were calculated for each analyzed tooth. The percentage of surfaces (facial and lingual) with BOP was also calculated and recorded as BOP%.

All unsalvageable teeth were extracted before periodontal treatment. After extraction of the unsalvageable teeth, part of the wall of the periodontal pocket was resected and harvested. The gingiva resected during the crown-lengthening surgery of the controls was also collected. Gingival samples from 17 subjects of the diseased group and twelve subjects of the control group were divided into part 1 and part 2. Gingival connective and epithelial tissues were obtained from part 1 using sharp tissue scissors, and then were stored in RNAwait (Solarbio Science & Technology Co., Beijing, China) at $-80\,°C$ until RNA extraction. Part 2 and gingival samples from the other subjects were dehydrated and embedded in paraffin and serial sections were cut with the microtome set at 5 $\mu$m. One section of each sample was examined after staining with hematoxylin and eosin (H&E).

### Detection of CYP27B1 expression

RNA was extracted using Trizol (Invitrogen, Carlsbad, CA, US) and was reverse transcribed to cDNA using a reverse transcription kit (TOYOBO Life Science (Shanghai), Shanghai, China). Real-time PCR reactions were performed using Faststart Universal SYBR Green Master Mix (Roche, Basel, Switzerland) in a real-time Thermocycler (Applied Biosystems, Foster City, CA, US) in triplicate. $\beta$-actin was used as an internal control (Forward primer: 5$'$-GCCGTGGTGGTGAAGCTGT-3$'$ and reverse

primer: 5′-ACCCACACTGTGCCCATCTA-3′). The forward primer for detection of CYP27B1 was 5′-ACGGTGTCCAACACGCTCT-3′ and the reverse primer was 5′-AACAGTGGCTGAGGGGTAGG-3′. Data are presented as relative mRNA levels calculated by the $2^{-\Delta Ct}$($\Delta Ct = Ct$ of target gene minus Ct of $\beta$-actin) (*Livak & Schmittgen, 2001*).

## Immunohistochemistry

Immunohistochemistry was performed according to previously described methods (*Li et al., 2017*). Briefly, selected sections were transferred onto adhesive slides (Zhongshan Golden Bridge Biotechnology Co., Beijing, China), deparaffinized with xylene and rehydrated with descending concentrations of ethanol, then digested with 1 g/L trypsin at 37 °C for 10 min for antigen retrieval. Endogenous peroxidase blocking was achieved by treatment with 3% $H_2O_2$ for 10 min at room temperature, then sections were incubated with primary sheep polyclonal antibody against CYP27B1 (1:100; The Binding Site Ltd., Birmingham, UK) at 4 °C for 12 h. This was followed by incubation with an anti-sheep secondary antibody (1:500; EarthOx Life Sciences, San Francisco, CA, US) at 37 °C for 30 min. The PV-9000 Polymer Detection System and a diaminobenzidine (DAB) kit (both from Zhongshan Golden Bridge Biotechnology Co., Beijing, China) were used for immunohistochemical staining of CYP27B1. The DAB staining time was 150 s for each section. Finally, sections were counterstained with hematoxylin.

## Immunofluorescence

After deparaffinization of the sections, antigen retrieval was accomplished by boiling in citric acid–sodium citrate buffer (0.01 M, pH 6.0) for 15 min, and endogenous peroxidase blocking was performed using the same method as described above. Then, sections were incubated with a working solution of primary rabbit anti-vimentin monoclonal antibody (ZA-0511; Zhongshan Golden Bridge Biotechnology Co., Beijing, China) at 4 °C for 12 h. Next, sections were incubated with an anti-rabbit secondary antibody (ZF-0511, diluted 1:400; Zhongshan Golden Bridge Biotechnology Co., Beijing, China) at 37 °C for 1 h. Then sections were incubated with primary mouse monoclonal antibody against CYP27B1 (sc-515903, diluted 1:50; Santa Cruz Biotechnology, Santa Cruz, CA, US) at 4 °C for another 12 h. The anti-mouse secondary antibody (sc-516141, diluted 1:50; Santa Cruz Biotechnology, CA, US) was added and incubated at 37 °C for 1 h, then nuclei were counterstained with DAPI (Neobioscience Biological Technology Co., Shenzhen, China), and sections were observed using immunofluorescence microscopy (Nikon, Tokyo, Japan).

## Image analysis

Image evaluation of the immunohistochemical results was performed by two experienced pathologists, who were unaware to which group the histological sections belonged. The CYP27B1 staining of each sample was rated as one of the following four grades: negative (-), weak (+), moderate (++) or strong (+++), translated as 0, 1, 2 and 3 for statistical analysis, respectively. The staining intensity of each sample was the consensus of the opinions of the two pathologists.

Each pathologist chose five non-overlapping 40× microscope fields of each section for evaluation of hGFs. The total number of hGFs and the number of immunohistochemically

CYP27B1-positive ones in each chosen microscope field were recorded and staining of gingival fibroblasts were rated as negative (-), weak (+), moderate (++) or strong (+++). The percentage of +, ++ or +++ hGFs was calculated and CYP27B1 staining score was calculated using the following formula: CYP27B1 staining score = (percentage of +++ cells) × 3 + (percentage of ++ cells) × 2 + (percentage of + cells). The method for calculating CYP27B1 staining score was previously reported by *Yoon et al. (2010)* and a higher score indicated stronger staining intensity. Cell counting and rating were performed by the two pathologists using a multi-person sharing microscope (Olympus, Tokyo, Japan) at the same time. They calculated the average CYP27B1 staining score of each section, and the mean of the two average CYP27B1 staining scores for each section was used for analysis.

Since immunofluorescence was usually used for qualitative analysis, the results of immunofluorescence staining were only used for the colocalization of vimentin and CYP27B1 in hGFs.

### Statistical methods

The Mann–Whitney U Test was used to compare AL, BI, BOP%, the relative mRNA expression of CYP27B1 in gingival epithelia and the staining grades of the two groups since normal distribution was not assumed. All the other comparisons between the two groups were carried out using Independent-samples T Test. Statistical analyses were carried out using the SPSS 11.5 software package (SPSS Inc., Chicago, IL, US). A $P$ value < 0.05 was considered statistically significant.

All the parameters were used to calculate power values using PASS 2008 (NCSS Inc., Kaysville, UT, US) and each power value was over 0.99.

## RESULTS

The characteristics of the two groups are shown in Table 1. Significantly higher PD and AL were observed in the periodontitis group than in the control group. BI and BOP% of all teeth analyzed in the periodontitis group were 4 and 100%, respectively. BI of the teeth analyzed in the control group was 0 or 1, so none had BOP.

The mRNA expressions of CYP27B1 in gingival connective tissues of patients with periodontitis were significantly higher than those in the gingival connective tissues of controls (Fig. 1A). In contrast, there was no significant difference in CYP27B1 mRNA expression between the gingival epithelia of the two groups (Fig. 1B).

The gingiva of one patient with periodontitis (Figs. 2A–2C) and one control (Figs. 2D–2F) are shown. Negative controls are shown in Figs. 2G–2I. The black frame indicates the epithelial tissue, and the blue frame indicates the connective tissue. As shown in the figure, the gingival connective tissues were CYP27B1 positive, and the expression of CYP27B1 was also detected in gingival epithelia. In the periodontitis group, the expression of CYP27B1 was detected in all epithelial layers, but expression was stronger in the superficial layer than in the deep layer of the epithelia in the control group. As shown in Figs. 3A–3F, the expressions of CYP27B1 and vimentin were colocalized, indicating that in gingival connective tissues, the cells positive for CYP27B1 expression were hGFs. Statistical analysis indicated that CYP27B1 staining intensities of the gingiva of patients with periodontitis

**Table 1  Demographic data and clinical parameters of the two groups.**

| Parameters | Periodontitis ($n = 42$) | Controls ($n = 33$) |
|---|---|---|
| Age (years) | 33.5 ± 7.8 | 30.0 ± 9.2 |
| Gender (male/female) | 20/22 | 16/17 |
| PD (mm) | 7.3 ± 0.4[a] | 1.9 ± 0.5 |
| AL (mm) | 5.9 ± 0.6[a] | 0 |
| BI | 4[a] | 0 (0 to 0.5) |
| BOP% | 100%[a] | 0 |

**Notes.**

Data are presented as mean ± SD, median (lower to upper quartile), or number of subjects, as indicated.

[a]Compared to the control group ($P < 0.05$).

[3.00, (3.00 to 3.00)] were significantly higher than those of the controls [1.00, (1.00 to 2.00)] (Fig. 4A). In all the 40× microscopic fields chosen for analysis, almost 100% of the hGFs were CYP27B1 positive and the CYP27B1 staining scores of hGFs of patients with periodontitis (2.49 ±0.08) were significantly higher than those of controls (1.84 ±0.12) (Fig. 4B).

## DISCUSSION

Our previous experiments in vitro verified that CYP27B1 is expressed in hGFs and that expression is up-regulated by the inflammatory stimuli, IL-1$\beta$ and sodium butyrate (*Liu, Meng & Hou, 2012b*). In the present study, we demonstrated the expression of CYP27B1 in gingival connective tissues in vivo. Since there are several types of cells in gingival connective tissues, immunofluorescence experiments were performed and demonstrated that CYP27B1 colocalized with vimentin, a marker of fibroblasts, indicating that the CYP27B1-positive cells in gingival connective tissues were hGFs. However, it should be pointed out that endothelial cells also express vimentin (*Piera-Velazquez & Jimenez, 2019*), as shown in Fig. 3. Additionally, as shown in Figs. 2 and 3, endothelial cells were also found to be CYP27B1 positive, which was in line with the results of *Zehnder et al. (2002)*. Because hGFs and endothelial cells can easily be distinguished by pathologists, our morphological analysis of hGFs was not influenced by endothelial cells. Although the actual situation in vivo is much more complex than that simulated in the laboratory, our observations that patients with periodontitis had higher mRNA expression of CYP27B1 and higher CYP27B1 staining scores were in line with our results in vitro. Therefore, our hypothesis that "CYP27B1 is expressed by hGFs in vivo and the expression might be positively associated with periodontal inflammation" was verified.

Although we focused on hGFs, epithelial tissues were also observed in the present study, and gingival epithelia were found to be CYP27B1 positive in both groups, which was in line with the findings of other researchers (*McMahon et al., 2011*; *Menzel et al., 2019*). What should be pointed out is that the distribution of CYP27B1 expression differed between the two groups (Fig. 3). When obtaining the gingival epithelial tissues for analysis of mRNA expression, it was impossible to obtain the entire epithelium clinically. Therefore, only the superficial layer was obtained to avoid contamination of connective tissues. Since

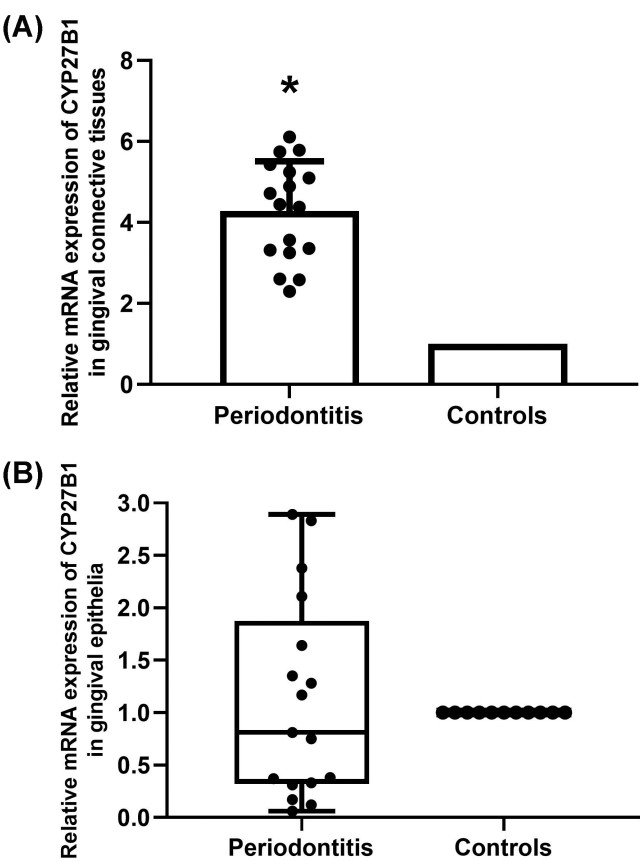

**Figure 1** **Expression of CYP27B1 mRNA in gingiva.** (A) Expression of CYP27B1 mRNA in gingival connective tissues of the diseased group was significantly higher than in the control group. (B) Expression of CYP27B1 mRNA in gingival epithelia did not significantly differ between the two groups.* Compared to the control group, $P < 0.05$.

CYP27B1 expression was relatively strong in the superficial layers in both groups, the lack of a significant difference in CYP27B1 expression between the epithelia of the two groups could be explained.

It has been demonstrated that a vitamin D pathway exists in hGFs (*Gao, Liu & Meng, 2018*) and hGEs (*McMahon et al., 2011*; *Menzel et al., 2019*). The pathway might be involved in periodontal immune defense for the following reasons. (1) 25OHD$_3$ alleviates experimental periodontitis in diabetic mice via the vitamin D pathway (*Zhou et al., 2018*). (2) In hGFs, the pathway is activated by the periodontal inflammatory stimulus *Pg*-LPS, and suppresses the expression of some inflammatory chemokines such as IL-8 and MCP-1 (*Gao, Liu & Meng, 2018*), indicating that the pathway might play a role in immune defense in periodontal soft tissues. (3) 25OHD$_3$ is an important part of the vitamin D pathway, and higher 25OHD$_3$ concentrations were detected in both the gingival crevicular fluids and the plasma of aggressive periodontitis patients compared to those of healthy controls (*Liu et al., 2009*). Moreover, after periodontal inflammation is reduced by initial periodontal therapy, the 25OHD$_3$ levels in gingival crevicular fluids and plasma significantly drop

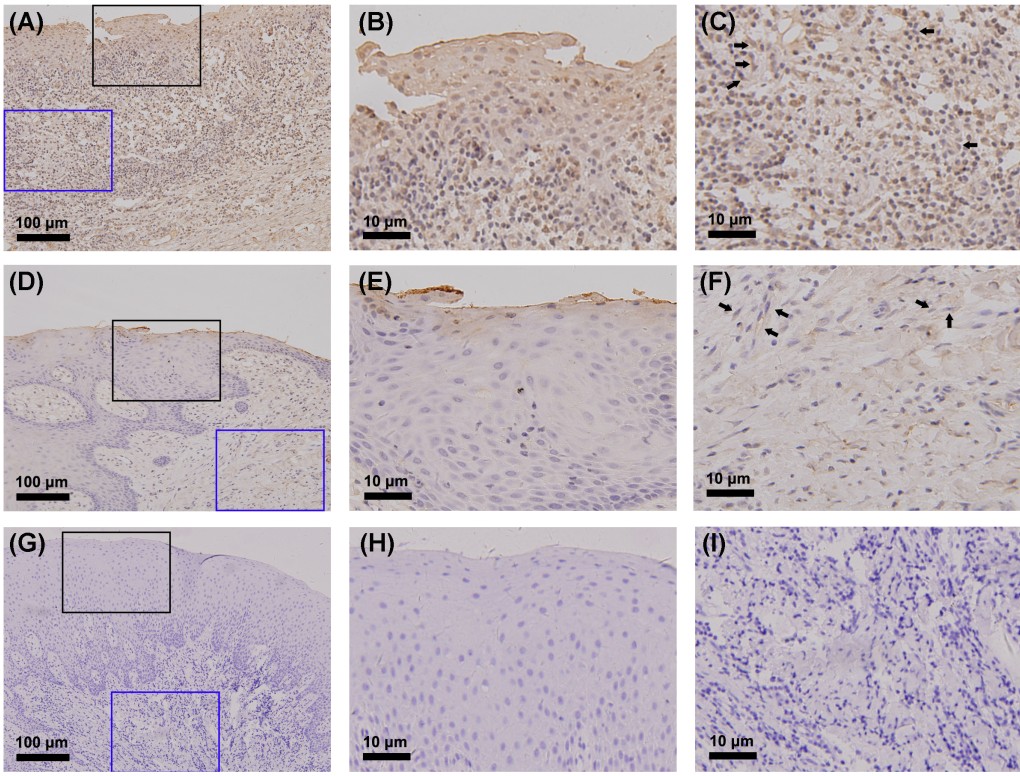

**Figure 2  Detection of CYP27B1 in gingiva by immunohistochemistry.** (A)–(C) and (D)–(F) show immunohistochemical staining of CYP27B1 in the gingiva of one patient with periodontitis and one control, respectively. (G)–(I) are negative controls. The black frame indicates the epithelial tissue (B, E, H), while the blue frame shows the connective tissue (C, F, I). The expression of CYP27B1 in hGFs is shown by arrows in (C and F) Magnification: (A), (D) and (G) 20×; all others 180×.

(*Liu et al., 2010*), indicating that activity of the vitamin D pathway might be positively associated with periodontal inflammation. As the key factor in the pathway (*Gao, Liu & Meng, 2018*), CYP27B1 is worthy of further research. In the present study, the finding that the in vivo gingival CYP27B1 expression was higher in the periodontitis group than in the control group could provide new evidence of the involvement of the vitamin D pathway in periodontal immune defense. According to the study by Tada et al. (*Tada et al., 2016*), $1,25OH_2D_3$ stimulation resulted in over 70-fold up-regulation of hCAP-18/LL-37 in an hGE cell line (Ca9-22). In contrast, $25OHD_3$ or $1,25OH_2D_3$ stimulation only resulted in 3- to 4-fold enhancement of expression of hCAP-18/LL-37 in hGFs (*Gao, Liu & Meng, 2018*). As the forefront of periodontal immune defense, it is reasonable that hGEs had a more active vitamin D pathway than hGFs. However, the relatively less active vitamin D pathway in hGFs is still worth studying, because once hGE, as the first line of periodontal defense, is breached and periodontal inflammation exacerbates to include gingival connective tissues, hGFs could play their role in periodontal immune defense through the vitamin D pathway. Thus, the present study is of biological significance, although more mechanisms via which the vitamin D pathway impacts gingival health in periodontitis still need to be elucidated.

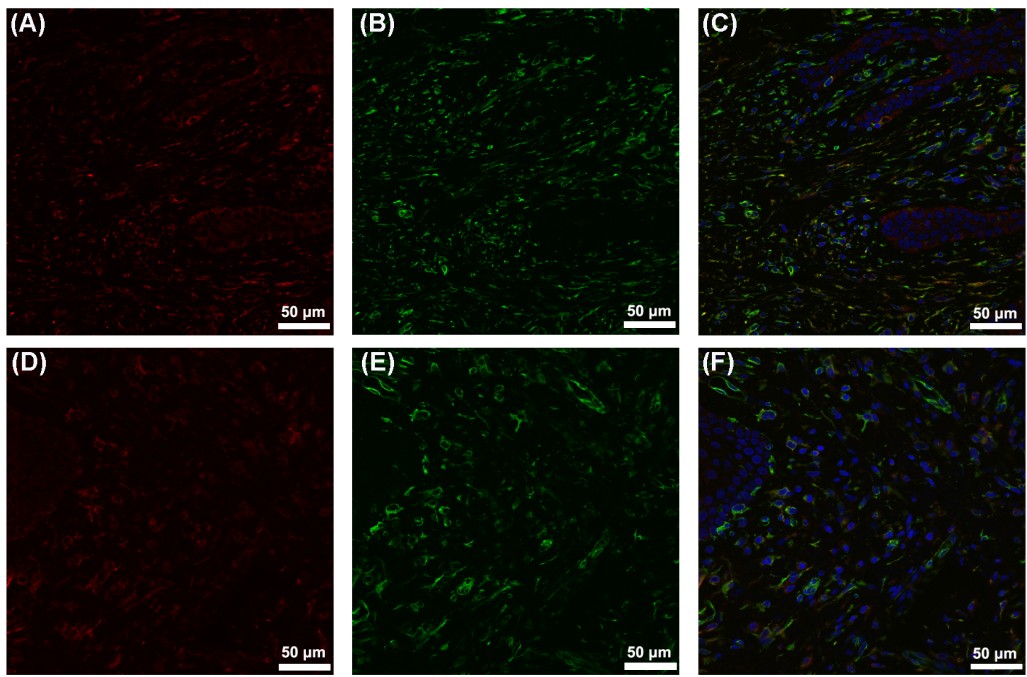

**Figure 3** **Colocalization of CYP27B1 and vimentin in hGFs.** (A)–(C) show immunofluorescence staining of samples from a periodontitis patient, and (D)–(F) show the corresponding results from a control. (A) and (D): CYP27B1; (B) and (E): vimentin; (C) and (F): combination.

Reasons for the higher expression of CYP27B1 in the periodontitis group might be as follows: (1) 25OHD$_3$ is an up-regulator of CYP27B1 in hGFs (*Gao, Liu & Meng, 2018*), and 25OHD$_3$ levels in gingival crevicular fluids of patients with periodontitis before initial periodontal therapy were significantly higher than those after therapy (*Liu et al., 2010*); (2) Periodontal inflammation results in higher concentrations of IL-1 $\beta$ and butyric acid in gingival crevicular fluids (*Liu et al., 2010*; *Lu et al., 2014*), which also induces the expression of CYP27B1 in hGFs (*Liu, Meng & Hou, 2012b*).

Our previous studies (*Liu et al., 2009*; *Liu et al., 2010*) indicated that systemic and local 25OHD$_3$ levels in patients with aggressive periodontitis were positively associated with periodontal inflammation. However, several existing studies (*Dietrich et al., 2004*; *Jimenez et al., 2014*; *Zhan et al., 2014*) suggested that vitamin D deficiency is associated with an increased risk of periodontal disease. What should be pointed out is that in these studies, the participants were about 50 years of age or older, an age range that did not overlap with that of the population in our previous studies (younger than 30 years old). Additionally, no correlation between plasma 25OHD$_3$ levels and periodontal health was found in another large cross-sectional study (*Antonoglou et al., 2015*), and the participants in that study were 30-49 years old. Thus, the relationship between 25OHD$_3$ and periodontitis in people of different ages might be different.

In studies investigating the association between 25OHD$_3$ and periodontal health in large samples (*Dietrich et al., 2004*; *Jimenez et al., 2014*; *Zhan et al., 2014*; *Antonoglou et al.,*

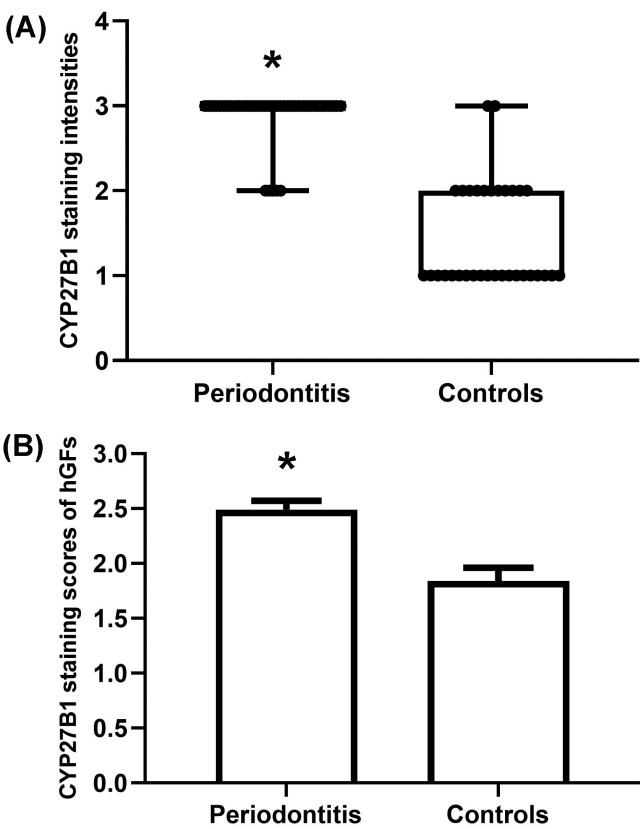

**Figure 4 Evaluation of CYP27B1 protein expressions in gingiva.** (A) CYP27B1 staining intensities of gingival connective tissues of the diseased group were significantly higher than those of the control group. (B) CYP27B1 staining scores of gingival fibroblasts of the diseased group were significantly higher than those of the control group.* Compared to the control group, $P < 0.05$.

*2015*), the participants were from the general population. However, in our previous study (*Liu et al., 2009*), only patients with aggressive periodontitis had higher plasma $25OHD_3$ levels, and the patients had much more severe periodontal inflammation than the other participants. In the special group, it is unclear whether the higher plasma $25OHD_3$ level is the reason for or the result of severe periodontal inflammation. Our previous study (*Gao, Liu & Meng, 2018*) suggested that $25OHD_3$ activates the vitamin D pathway, which participates in periodontal immune defense. Therefore, it is possible that, due to severe periodontal inflammation, more LL37 is needed for antibacterial and anti-inflammatory function, and more $25OHD_3$ is synthesized for the more active vitamin D pathway in periodontium. This possibility could help to explain why patients with severe periodontitis had higher systemic and local 25OHD3 levels. In the present study, the results that patients with periodontitis had higher CYP27B1 expression in hGFs indicated that patients with periodontitis had a more active vitamin D pathway in hGFs, which further supported this possibility.In the present study, subjects matched by age and gender were included in the two groups and all were non-smokers, in order to minimize the influence of potential confounding factors. To analyze the typical inflammatory situation in vivo, all

the patients enrolled were diagnosed with periodontitis Stage IV Grade C, the most severe periodontitis in the new classification scheme for periodontal diseases (*Tonetti, Greenwell & Kornman, 2018*; *Papapanou et al., 2018*). In addition, all gingival tissues of patients with periodontitis were obtained around unsalvageable teeth, which had not received any periodontal therapy so that periodontal inflammation was serious enough and was not influenced by periodontal treatments. The PD and AL of the unsalvageable teeth analyzed were high and BOP was positive at all surfaces of the teeth. In contrast, all the teeth analyzed in the control group had PD less than three mm and had no AL or BOP, indicating that these teeth were clinically healthy. It should be pointed out that all the teeth analyzed in the control group needed crown-lengthening surgery because of excessive gingival display or subgingival location of fracture lines or carious lesions. When parts of the teeth were subgingival, accumulation of dental plaque was often detected. Thus, the BI of some teeth in the control group was 1 and mild inflammation of the gingiva could be detected. Similarly, it was reported that "healthy" gingiva might also harbor inflammatory cellular infiltrates, indicating that subclinical gingivitis might exist (*Lang & Bartold, 2018*). Thus, CYP27B1 staining intensities of two of the 33 teeth in the control group were strong (+++) and the mild inflammation of the gingiva might be the reason for high expression of CYP27B1 in the control group.

Immunohistochemistry is of course a highly subjective method. We tried to objectively evaluate CYP27B1 expression in gingiva in vivo by letting two experienced pathologists perform the evaluation in a blinded manner. However, the subjectivity of the evaluation was inevitable, which is a limitation of the present study.

## CONCLUSIONS

In summary, CYP27B1 expression was detected in hGFs in vivo, and this expression might be induced by periodontal inflammation. These results validated our previous in vitro findings, and indicated the potential involvement of the vitamin D pathway in periodontal immune defense. The present study can help lay the foundation for using vitamin D pathway in the treatment of periodontitis via vitamin D supplement.

### Funding
This work was supported by the National Natural Science Foundations of China (No. 81100749, 81271149). The funders had no role in study design, data collection and analysis, decision to publish, or preparation of the manuscript.

### Grant Disclosures
The following grant information was disclosed by the authors:
National Natural Science Foundations of China: 81100749, 81271149.

### Competing Interests
The authors declare there are no competing interests.

## Author Contributions

- Kaining Liu conceived and designed the experiments, performed the experiments, prepared figures and/or tables, authored or reviewed drafts of the paper, and approved the final draft.
- Bing Han performed the experiments, prepared figures and/or tables, authored or reviewed drafts of the paper, and approved the final draft.
- Jianxia Hou conceived and designed the experiments, performed the experiments, authored or reviewed drafts of the paper, and approved the final draft.
- Jianyun Zhang and Jing Su analyzed the data, authored or reviewed drafts of the paper, and approved the final draft.
- Huanxin Meng conceived and designed the experiments, analyzed the data, authored or reviewed drafts of the paper, and approved the final draft.

## Human Ethics

The following information was supplied relating to ethical approvals (i.e., approving body and any reference numbers):

The institutional review board of Peking University School and Hospital of Stomatology approved the study protocol (PKUSSIRB-2011007).

## Data Availability

Data is available at FIgshare: Liu, Kaining; Han, Bing; Hou, Jianxia; Zhang, Jianyun; Su, Jing; Meng, Huanxin (2020): Expression of vitamin D $1\alpha$-hydroxylase in human gingival fibroblasts in vivo. figshare. Dataset. https://doi.org/10.6084/m9.figshare.12463874.v1

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
