# Peer review of "Expression of vitamin D 1α-hydroxylase in human gingival fibroblasts in vivo"

_PeerJ, doi:10.7717/peerj.10279_

## Round 0.1 · original submission · Major Revisions

Please address all the concerns, especially those from reviewer 2.

·

Basic reporting

The article is clearly written, providing sufficient literature references for background, methodology and discussion. The structure of the article is appropriate with figures and tables properly designed.

One issue is that the data in figures 1 and 2 are presented as relative mRNA expression, apparently each is relative to b-actin levels. However, figure 2 is simply a reshowing of figure 1. Is this just the controls or the periodontitis? Regardless, it would be much more appropriate to make the control set to 1.0 for all cases in figure 1, with the periodontitis shown relative to that amount. Then, it would make sense to show the data in figure 2, setting either one as the 1.0. As it stands, there is no reason for figure 2, and the data are hard to understand in figure 1.

Experimental design

The authors have previously shown that CYP27B1 is expressed in vitro, so it is important to show this expression in vivo, especially where it is difficult to distinguish cell type (fibroblast vs. epithelium). Thus, the immunohistochemistry data are important. The results are clearly shown and support the in vitro data. All of the methods are sufficiently detailed. That said, the ultimate significance of the fact that CYP27B1 is expressed in these tissues is not well elaborated. It would strengthen the manuscript to explain to the reader why it is important to know A) that CYP27B1 is expressed in both (or either) cell type, and B) that CYP27B1 expression apparently differs in hGF vs. hGE in periodontal disease.

Validity of the findings

The data appear to be fairly well presented. It would strengthen the rigor of the study to present the PCR data with each data point (either as a box plot or to keep the column graph, but to indicate each data point). The conclusions are presented, but some speculation on the further utility of this information would truly strengthen the manuscript.

Additional comments

Overall, this is a straightforward study demonstrating the expression of CYP27B1 in gingival tissue in vivo. It is important to the field to know this information, but the overall significance of the findings should be elaborated upon.

Reviewer 2 ·

Basic reporting

Literature analysis is incomplete, own works are over cited.
Some figures need to be improved.

Experimental design

There are some questions regarding patient selection and tissue sampling.

Validity of the findings

The number of replicates is not mentioned for some quantitative data.

Additional comments

In the present study, the expression of CYP27B1 in the gingival soft tissue of periodontitis patients and healthy controls is compared. Vitamin D3 seems to play an essential role in periodontitis. Several previous studies of this group and other groups show that vitamin D3 could be locally converted into 25(OH)D3 and subsequently into bioactive 1,25(OH)2D3 by several cells of periodontal tissue.

The primary concern is the sampling of gingival tissue. In periodontitis patients, samples were taken from “part of the wall of periodontal pocket”, whereas, in the healthy subjects, the gingiva resected during crown lengthening procedure was used. First, how these anatomically different tissues could be compared? Second, the wall of the periodontal pocket could be the alveolar bone.

The second critical point is the control of serum vitamin D level or at least vitamin D supplementation by the study participants.

Further comments
Lines. 58-61. The conversion of vitamin D3 into 25(OH)D3 by the liver and subsequently into 1,25(OH)2D3 by the kidney is well known. However, the studies mentioned here (Liu et al. 2012a,b) refers only to the vitamin D3 conversion by gingival fibroblasts/periodontal ligament cells.

Line 62. 1-alpha hydroxylase CYP27B1 expression in the kidney was first detected before the study of Nykjaer et al.

Line 85. No analysis of CYP27B1 expression in endothelial cells was performed. Endothelial cells should be counterstained with CD31 or von Willebrand factor. Periodontal inflammation is characterized by increased vascularization and, therefore, the increased number of endothelial cells. Thus, the possibility that the increased CYP27B1 expression is due to higher endothelial cell numbers should be excluded.

The number of replicates should be shown for each experiment.
Gene expression analysis was done only for some 17 periodontitis patients and 12 healthy controls. How were these patients selected? What were the demographic and clinical data of these patients?
Immunohistochemistry, Figure 3. The expression of CYP27B1 should be shown by arrows so that inexperienced readers can recognize it.
Figure 3. Scale bars should be used instead of providing the magnifications because the pictures could be scaled during the publication process.
Figure 4. Any scale bar is absent.
Fig 5A. What kind of plot is it? Is it a box-plot? In this case, the median and quartiles should be visible. Otherwise, some graphical presentations, for example, histogram, should be used.

Discussion; lines 246-255. How could the increased level of 25(OH)D3 in the gingival crevicular fluid be related to the CYP27B1 expression? CYP 27B1 converts 25(OH)D3 into biologically active 1,25(OH)2D3 but is hardly connected to the local 25(OH)D3 levels.

Most existing literature suggests that vitamin D deficiency is associated with an increased risk of periodontal disease. This fact should be discussed in connection with the obtained data.

---

## Round 0.2 · accepted · Accept

Dear Dr Liu,

Thanks for addressing all concerns by the reviewers.

·

Basic reporting

no comment

Experimental design

no comment

Validity of the findings

no comment

Additional comments

The authors have appropriately responded to the initial review.

Reviewer 2 ·

Basic reporting

No comment

Experimental design

No comment

Validity of the findings

No comment

Additional comments

Thank you for considering the criticism.